# The dicarbon bonding puzzle viewed with photoelectron imaging

B.A. Laws [1]*, S.T. Gibson [1], B.R. Lewis[1] & R.W. Field [2]

Bonding in the ground state of $C_2$ is still a matter of controversy, as reasonable arguments may be made for a dicarbon bond order of 2, 3, or 4. Here we report on photoelectron spectra of the $C_2^-$ anion, measured at a range of wavelengths using a high-resolution photoelectron imaging spectrometer, which reveal both the ground $X^1\Sigma_g^+$ and first-excited $a^3\Pi_u$ electronic states. These measurements yield electron angular anisotropies that identify the character of two orbitals: the diffuse detachment orbital of the anion and the highest occupied molecular orbital of the neutral. This work indicates that electron detachment occurs from predominantly $s$-like ($3\sigma_g$) and $p$-like ($1\pi_u$) orbitals, respectively, which is inconsistent with the predictions required for the high bond-order models of strongly $sp$-mixed orbitals. This result suggests that the dominant contribution to the dicarbon bonding involves a double-bonded configuration, with $2\pi$ bonds and no accompanying $\sigma$ bond.

[1] Research School of Physics and Engineering, The Australian National University, Canberra, ACT 2601, Australia. [2] Department of Chemistry, Massachusetts Institute of Technology, Cambridge, MA 02139, USA. *email: Ben.Laws@anu.edu.au

Despite the relative simplicity of homonuclear diatomic molecules, the bonding nature of dicarbon, $C_2$, has long been a topic of debate. This discussion has been driven by recent advances in computational methods, with various studies suggesting the carbon–carbon bond may have a bond order of 2, 3, or even 4, with the latter from ab-initio studies[1]. Standard qualitative theories predict different values for the dicarbon bond order[2,3]. From a simple Lewis electron-pair repulsion description, the 8 valence electrons in $C_2$ are predicted to form a quadruple bond. However, this bonding assignment seems unlikely, with stable quadruple bonds typically only found between transition metal elements that have partially filled $d$-orbitals[4]. In contrast, a hybrid-orbital (HO) style approach invokes $sp^1$ hybridisation to predict a triple bond between the carbons. However, if molecular orbital (MO) theory is used, a ground-state valence electron configuration for $C_2$ of $KK(\sigma_{2s})^2(\sigma_{2s}^*)^2(\pi_{2p})^4$ is predicted, yielding a bond order of 2, with the unusual situation of a $\pi$ double bond with no accompanying $\sigma$ bond.

From an ab-initio approach, standard Hartree–Fock-based calculations support a dicarbon double bond; however, more advanced theoretical studies have suggested that the C–C bond may be better described by a higher bond order[5–8]. A recent high-level full configuration–interaction calculation, combined with valence bond (VB) theory, identified four distinct contributions to the bonding in $C_2$[1]. This included contributions from a $\sigma$ bond, $2 \times \pi$ bonds, plus an interaction between the outward pointing $sp^1$ hybrid orbitals[1]. The strength of this inverted bond between the $sp^1$ orbitals has since been calculated at various levels of theory and is estimated to contribute $\sim$50–80 k mol$^{-1}$[1,5,9,10]. This high bond-order model is further supported by a subsequent quantum chemistry calculation, which found higher magnetic shielding in $C_2$ compared with $C_2H_2$, supporting a bulkier C–C bond[11]. However, not all of the recent studies are in agreement, with some research preferring the notion of a double[12], triple[3], or quasi double–triple[13] bond, whereas other studies note that the theoretical approaches are not sufficient to definitively discern between the different bonding models[14–16].

Despite advances in spectroscopic techniques, experimental studies have not been able to confirm any of the suggested bonding structures, with the majority of the debate currently driven by the results of ab-initio calculations[2]. Due to the highly reactive nature of $C_2$, most experimental studies involve flame-emission[17–20] or plasma-discharge[21,22] spectroscopy. These studies support a bond order between 2 and 3, with a measured C–C bond length of 1.243 Å, longer then a typical alkyne triple bond, but shorter then a typical alkene double bond[19]. Likewise, a measured C–C bond dissociation energy of 602 kJ mol$^{-1}$, as well as a calculated bond restoring force of 12 N[14], also lie between double and triple bond limits[23].

As the dicarbon anion $C_2^-$ is stable, photoelectron spectroscopy may be used to probe the reactive $C_2$ neutral molecule[24,25]. The first dicarbon photoelectron experiment was performed by Ervin et al.[24], on $C_2^-$ anions produced in an $O^-$/HCCH afterglow ion source. This source produced hot anions, with multiple hot bands present in the spectrum, and defined an accurate value for the electron affinity of $C_2$ of 3.269(6) eV[24]. A later study by Bragg et al.[26] probed the low-lying excited states of $C_2$, in a single wavelength measurement at 264 nm (as part of a larger study, employing time-resolved photoelectron spectroscopy to examine transitions from excited anion states). This study provided the first experimental anisotropy measurement of photodetachment from the $C_2^-$ anion.

In this work, the photoelectron spectrum of $C_2^-$ is revisited using a high-resolution photoelectron imaging (HR-PEI) spectrometer. Although it is now well established that the photoelectron angular distributions indicate the character of the relevant molecular anion orbitals[27], electric dipole selection rules also influence the electron anisotropy via the character of the final neutral states, providing insight to understand the nature of the bonding between carbon atoms in $C_2$. Here we report on photoelectron angular distributions measured at a range (290 to 355 nm) of wavelengths, to probe the character of the $C_2$ orbitals. This reveals detachment occurs from pure $s$-like ($3\sigma_g$) and $p$-like ($1\pi_u$) orbitals, suggesting that the dominant contribution to the dicarbon bonding involves a double-bonded configuration, with $2\pi$ bonds and no accompanying $\sigma$ bond.

## Results and Discussion

**Theoretical discrepancies.** The simple idea of a bond order, defined as

$$\frac{\text{Bonding electrons} - \text{Antibonding electrons}}{2} \quad (1)$$

is often a useful way for chemists to describe the bonding nature in molecules. However, this can be an oversimplification in systems where the contribution from different bonding/anti-bonding electrons are unequal. Furthermore, the concept breaks down entirely in systems, which are intrinsically multi-reference in nature. Thus, when discussing the bonding in a molecule as subtle as $C_2$, it is important to keep in mind that there are multiple configurations all contributing to the overall state.

Ab-initio descriptions of the bonding are also complicated by the multi-reference nature of $C_2$. Specifically, it is the quasi-degeneracy of the $2\sigma_u^*$, $1\pi_u$ and $3\sigma_g$ molecular orbitals, responsible for the numerous low-lying excited electronic states of $C_2$, which muddle the bonding picture. Much of this uncertainty is influenced by the nature of the $2\sigma_u^*$ orbital, which is predicted to be a very weakly anti-bonding orbital[1,28]. To account for this behaviour, an alternative orbital scheme has been proposed, involving hybridisation of the $2\sigma_u^*/3\sigma_g$ molecular orbitals, forming $sp^1$-like singly occupied hybrid orbitals ($2\sigma_u^* + 3\sigma_g$) and ($2\sigma_u^* - 3\sigma_g$)[5], as depicted in Fig. 1.

To visualise the differences between these possible bonding pictures, ab-initio calculations were performed in this work using NWChem[29] and Q-Chem[30] software. Most standard methods for approximately solving the Schrödinger equation for molecules are based on Hartree–Fock [or self-consistent field (SCF)] theory, utilising a mean-field approximation. However, for molecules with quasi-degenerate or low-lying excited states, a multi-configurational complete active space (CASSCF) approach may be required. This approach accounts for states that are a linear combination of several quasi-degenerate configurations, allowing for non-integer orbital occupation numbers.

The molecular orbitals for $C_2$ were calculated with both SCF and CASSCF methods, using a cc-pVTZ Dunning basis set[31], as shown in Fig. 2. The key difference between the two approaches is the ordering of the $2\sigma_u^*$ orbital, with the multiconfigurational calculation increasing the orbital energy due to possible mixing between the $2\sigma_u^*/3\sigma_u^*$ and $2\sigma_g/3\sigma_g$ orbitals. This mixing is also illustrated in the occupation numbers, with a substantial ($\sim$0.4) occupation found in the $3\sigma_g$ orbital. As the dominant weight in this mixed orbital lies with the anti-bonding $2\sigma_u^*$, these occupation numbers suggest a bond order of three, consistent with a pure hybrid-orbital theory argument[6]. However, if this $2\sigma_u^*/3\sigma_u^*$ and $2\sigma_g/3\sigma_g$ mixing is allowed to occur before the calculation, using hybrid orbitals $\varphi_L = 2\sigma_u + 3\sigma_g$ and $\varphi_R = 2\sigma_u - 3\sigma_g$, the predominant configuration is for each orbital to be singly occupied, with one electron on each of the carbons[5]. From Hund's Rule, one may expect the lowest energy

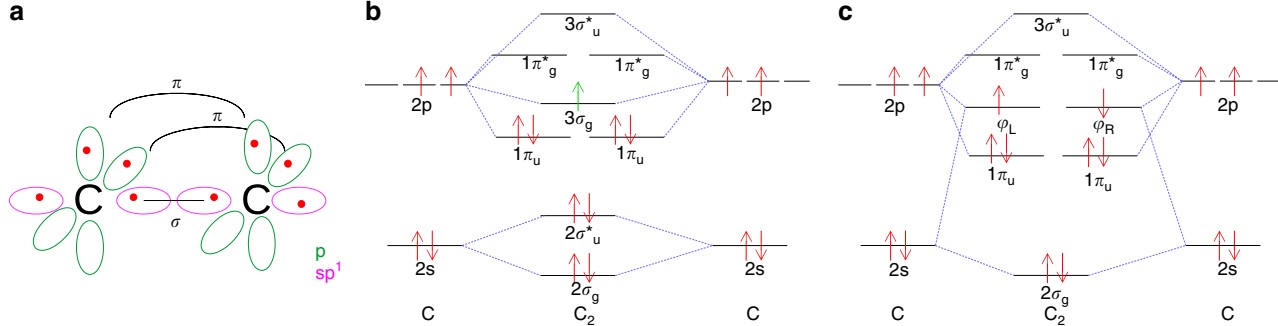

**Fig. 1** Orbital descriptions of $C_2$ using different approaches, highlighting the uncertain bond nature of dicarbon. **a** Hybrid-orbital picture of $C_2$, involving $sp^1$ hybridisation on the carbon atoms. This approach suggests a triple-bonded dicarbon molecule, with one $\sigma$ and $2\pi$ bonds, as represented by solid black lines. **b** Molecular orbital diagram for $C_2$, predicting a double bond between the C atoms consisting of $2\pi$ bonds, without an accompanying $\sigma$ bond. The additional electron for the anion $C_2^-$ is shown in green. **c** Orbital diagram for $C_2$ from a valence-bond viewpoint. Due to the quasi-degenerate nature of orbitals $2\sigma_u$, $3\sigma_g$ and $1\pi_u$, the $2\sigma_u^*$ and $3\sigma_g$ orbitals mix to form $sp^1$-like hybrid orbitals on each of the carbons. Valence bond theory suggests a fourth bond involving the interaction between the outward pointing $\varphi$ orbitals

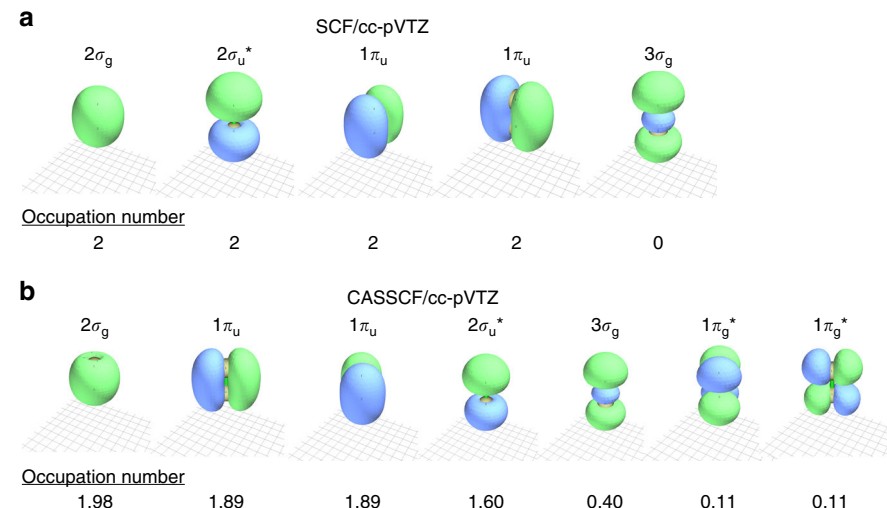

**Fig. 2** Valence molecular orbitals of $C_2$ calculated using NWChem software[29]. **a** Orbitals calculated using an SCF (self-consistent field) approach, recreating the molecular orbital diagram in Fig. 1b. **b** Orbitals calculated using a multiconfigurational CASSCF (complete active space self-consistent field) approach, which agree with the valance bond orbital diagram (Fig. 1c). Orbital occupation numbers are also given

configuration to be the triplet state, with the odd electrons occupying the degenerate hybrid orbitals in the same spin state. However, generalized valence bond calculations, using the principal of maximum coupling between overlapping atomic orbitals, have shown that the energy gain from the coupling interaction between the singlet-paired electrons in each hybrid orbital is greater than the energy cost from violating Hund's Rule[1,32]. This is consistent with the VB picture (Fig. 1c), used to suggest a dicarbon quadruple bond, whereas the SCF results are consistent with the double-bonded MO picture (Fig. 1b). Orbitals of the $C_2^-$ anion were also calculated using the same (cc-pVTZ) basis. The addition of an electron has a minimal impact on the structure of the inner orbitals, with a calculated overlap integral of the $\pi_u$ anion and neutral orbitals giving 85% similarity, with the 15% difference accounted for by the slight change in the $C_2$ bond length.

**High-resolution photoelectron imaging**. To investigate the uncertainty regarding the bonding nature of dicarbon, photoelectron spectra of $C_2^-$ were measured, using the HR-PEI spectrometer at the Australian National University. Dicarbon anions are produced in a single pulsed-jet discharge source of pure ethylene, with the subsequent ion mass separated by time-of-flight. Ions are then photodetached at 355 nm using the third harmonic of a Nd:YAG laser, with the electrons mapped onto a detector using velocity-map imaging (VMI). This allows for energetic and angular information to be measured simultaneously. A raw velocity-mapped image of $C_2^-$ photodetachment, corresponding to 414,511 electrons, is presented in Fig. 3. The image contains two rings, corresponding to the $C_2 X^1\Sigma_g^+ + e^- \leftarrow C_2^- X^2\Sigma_g^+ + h\nu$ and $C_2 a^3\Pi_u + e^- \leftarrow C_2^- X^2\Sigma_g^+ + h\nu$ photodetachment transitions. Because of the quasi-degeneracy of the $2\sigma_u^*$, $1\pi_u$ and $3\sigma_g$ molecular orbitals, the dicarbon neutral molecule has many low-lying electronic states, which leads to the unusual property whereby the term energy of the first excited state [718.318(2) cm$^{-1}$][33] is smaller than the vibrational frequency [$\omega_e = 1854.5(8)$ cm$^{-1}$][24]. Consequently, at a detachment wavelength of 355 nm we observe two electronic transitions, but only the vibrational original of each state.

Rotational band models for the $C_2(X^1\Sigma_g^+) \leftarrow C_2^-(X^2\Sigma_g^+)$ and $C_2(a^3\Pi_u) \leftarrow C_2^-(X^2\Sigma_g^+)$ electronic transitions have been constructed using methods similar to those developed by Buckingham et al.[34]. A detailed derivation of the molecular rotational model may be found in the Supplementary Discussion. The Hunds case-(b) to case-(b) and case-(b) to case-(a) rotational models were fitted the

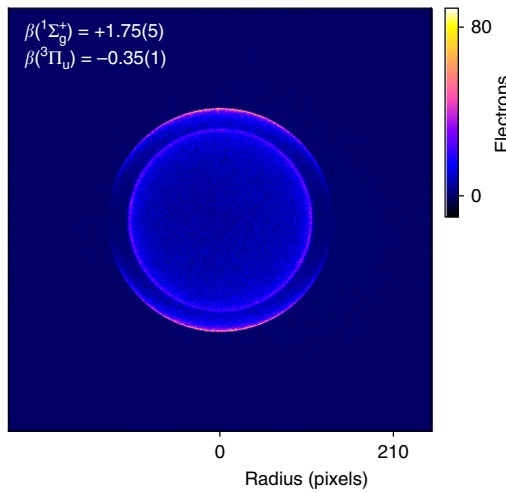

$\beta(^1\Sigma_g^+) = +1.75(5)$
$\beta(^3\Pi_u) = -0.35(1)$

**Fig. 3** Velocity-map image of $C_2^-$ at 355 nm. Fast electrons are mapped to the detector edge with slow electrons at the centre, whereas the laser polarisation defines the vertical axis. Two transitions are observed, with detachment to the ground $X^1\Sigma_g^+$ state observed as an outer ring and detachment to the first excited $a^3\Pi_u$ state represented by the inner ring. The two transitions display opposite anisotropies, with the $^1\Sigma_g^+$ electrons preferentially distributed at the poles of the image and the $a^3\Pi_u$ electrons weakly focused around the horizontal axis

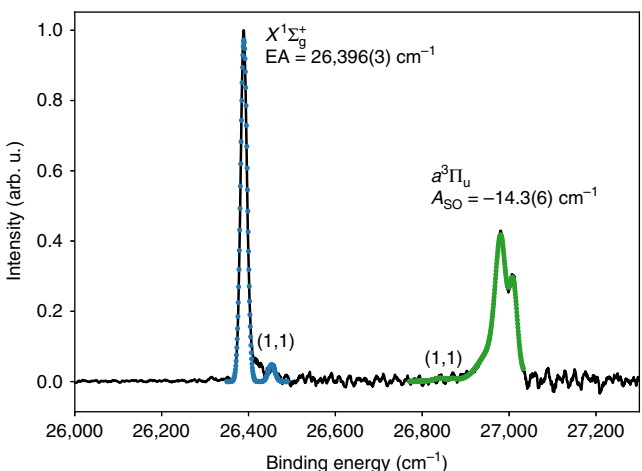

**Fig. 4** Photoelectron spectrum of $C_2^-$ at 355 nm. Two electronic transitions $C_2(X^1\Sigma_g^+) \leftarrow C_2^-(X^2\Sigma_g^+)$ and $C_2(a^3\Pi_u) \leftarrow C_2^-(X^2\Sigma_g^+)$ are observed, separated by 612 cm$^{-1}$. The experimental spectrum is shown in black, with a rotational band model ($X \leftarrow X$ shown in blue, $a \leftarrow X$ shown in green) fitted to each transition. This highlights the presence of a hot band (1, 1)

**Table 1 Franck–Condon factors for the vibrational transitions in $C_2^-$ photodetachment**

|       | $X^1\Sigma_g^+ \leftarrow X^2\Sigma_g^+$ | $a^3\Pi_u \leftarrow X^2\Sigma_g^+$ |
|-------|------|------|
| (0, 1) | 0.090 | 0.231 |
| (0, 0) | 0.902 | 0.756 |
| (1, 0) | 0.096 | 0.209 |
| (1, 1) | 0.726 | 0.365 |
| (2, 1) | 0.177 | 0.287 |
| (2, 2) | 0.198 | 0.089 |

validate the assignment of the $(1, 1)$ bands in Fig. 4, Franck–Condon factors were calculated. Potential energy curves of the anion $X^2\Sigma_g^+$ and neutral $X^1\Sigma_g^+$ and $a^3\Pi_u$ states were constructed using the Rydberg–Klein–Rees inversion method, with the resulting vibrational wavefunction overlap integrals presented in Table 1.

Table 1 confirms that photodetachment from vibrationally hot anions yielding $C_2(X^1\Sigma_g^+)$ will have a maximum intensity at $(1, 1)$, explaining the absence of the $(0, 1)$ hot band in the spectrum in Fig. 4. Photodetachment yielding $C_2(X^3\Pi_u)$ also has a maximum intensity at $(1, 1)$: however, this transition has a much smaller Franck–Condon factor than the corresponding transition in the ground state. Consequently, this transition has less intensity in the experimental spectrum. From the relative intensities of the $(0, 0)$ and $(1, 1)$ transitions in the photoelectron spectrum, the vibrational temperature of the $C_2^-$ anion may be defined. This analysis reveals a temperature of $T_{vib} \sim 900$ K, which is significantly higher than the rotational temperature derived from the model fit in Fig. 4 [$T_{rot} = 197(2)$K]. This is also higher than the normal operating conditions of the spectrometer, suggesting that the high-voltage discharge source preferentially produces dicarbon anions in vibrationally excited states.

The photoelectron spectrum in Fig. 4 may be calibrated using the term energy [$T_e = 718.318(1)$ cm$^{-1}$] of the excited $a^3\Pi_u$ state, which is well-defined from neutral flame-emission spectroscopy studies[35]. The rotational band model provides an accurate position for the band origin, allowing for the precise determination of molecular constants (listed in the Supplementary Table 1). From the calibration of the high-resolution photoelectron spectrum presented here, a precise value for the electron affinity of EA = 3.2727(4) eV is determined, in agreement with the previously accepted value of Lineberger et al.[24] of EA = 3.269(6) eV. The anion ground-state rotational [$B = 1.746(1)$ cm$^{-1}$] and vibrational [$\omega_e = 1782(2)$ cm$^{-1}$] constants are also extracted.

To investigate the relationship between the photoelectron spectrum of $C_2^-$ and the laser detachment energy, additional measurements were recorded by pumping a Sunlite Optical Parametric Oscillator (OPO) with the third harmonic of a Nd:Yag laser. To achieve photon energies greater than the electron affinity [3.2727(4) eV], the OPO output was doubled using a Continuum FX-Doubler. This allowed for photoelectron spectra of $C_2^-$ to be measured at a range of wavelengths (290–325 nm), as shown in Fig. 5. The OPO measurements show both the origin (0, 0) and first excited (1, 0) vibrational transitions for the ground $X^1\Sigma_g^+$ and excited $a^3\Pi_u$ electronic states.

**Bonding insights from the anisotropy.** The angular distribution of the photoelectrons, as measured by the VMI lens, is typically not isotropic and is related to the character of the parent anion molecular orbital. For detachment using linearly polarised light,

the ground ($X^1\Sigma_g^+$) and excited ($a^3\Pi_u$) photoelectron transitions, respectively, as shown in Fig. 4. The model fit yields an anion rotational temperature of $T = 197(2)$ K, whereas the Gaussian full-width half-maximum of each transition gives an energy resolution of $\Gamma_{X^1\Sigma} = 15(1)$ cm$^{-1}$ and $\Gamma_{a^3\Pi} = 18(1)$ cm$^{-1}$, which are consistent with expectations for the spectrometer at electron kinetic energies of $\epsilon \sim 0.3$ eV.

The rotational model also shows additional weak transitions to the right of the $X \leftarrow X$ peak and to the left of the $a \leftarrow X$ peak. These signatures are assigned to the $(1, 1)$ hot-band transition. In the earlier measurement of Ervin et al.[24], strong $(0, 1)$ hot-band transitions were observed; however, only the $(1, 1)$ transitions appear to be present in the photoelectron spectrum presented here, possibly due to a lower source temperature. To

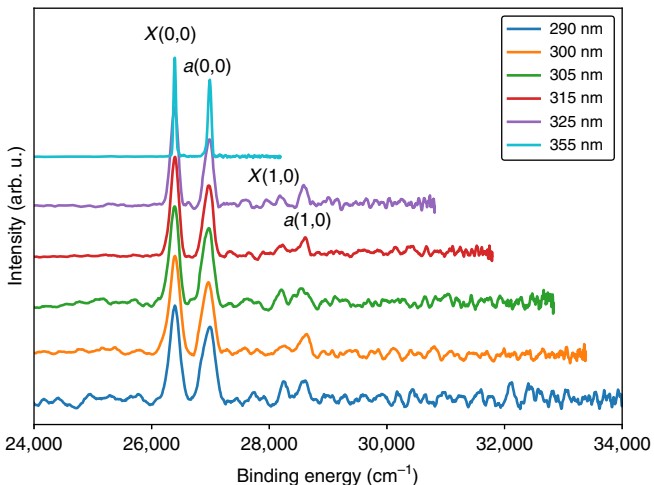

**Fig. 5** Photoelectron spectra of $C_2^-$ at a range (290–355 nm) of detachment wavelengths. The vibrational origin transitions are observed for the ground $X^1\Sigma_g^+$ and first excited $a^3\Pi_u$ electronic states. The OPO measurements (290–325 nm) also show the (1, 0) vibrational transition for each state

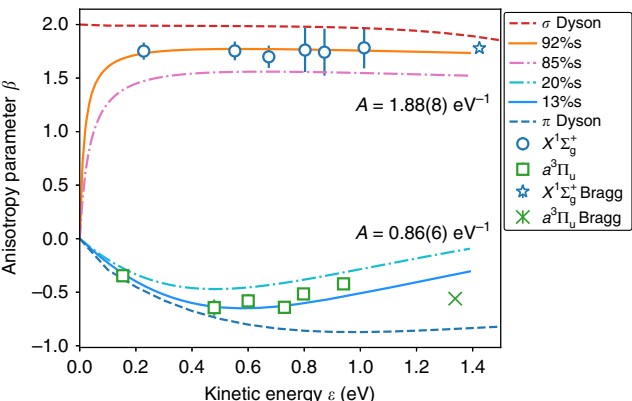

**Fig. 6** Anisotropy parameters for $C_2^-$ photodetachment. Experimental values are show for detachment at 295–355 nm [from this work (circles/squares)] and at 264 nm [from ref. [26] (stars/crosses)]. Detachment to the ground $X^1\Sigma_g^+$ state has a Hanstorp coefficient of $A = 1.88(8)$ eV$^{-1}$ with an orbital character of $f = 92\%\,s$ (orange curve), whereas detachment to the excited $a^3\Pi_u$ state has a Hanstorp coefficient of $A = 0.86(6)$ eV$^{-1}$ and an orbital character of $f = 13\%\,s$ (blue curve). Anisotropy curves calculated from Dyson orbitals are also shown (red and blue dashed lines). Error bars represent 1 SD, defined by the covariance matrix of the least-squares fit to Eq. (2)

the differential cross-section is given by

$$\frac{d\sigma}{d\Omega} = \frac{\sigma_{\text{total}}}{4\pi}[1 + \beta P_2(\cos\theta)], \qquad (2)$$

where $\theta$ is the angle between the ejected electron and the (vertical) laser polarisation, and $P_2$ is the second-order Legendre polynomial. The anisotropy parameter ($\beta$) may take values ranging from $-1$ to $2$, for purely perpendicular and parallel electronic transitions, respectively[34]. $\beta$ is a quantitative measure of how anisotropic the electron distribution is, with $\beta = 0$ corresponding to a perfectly isotropic distribution.

A qualitative description of the anisotropy parameter may be determined by visual inspection of the velocity-map image in Fig. 3. Noticeably, the two electronic transitions have different photoelectron angular distributions, with the electrons from detachment to $X^1\Sigma_g^+$ preferentially distributed at the poles of the image signalling a strong positive anisotropy parameter, whereas the $a^3\Pi_u$ electrons appear to have a slight negative anisotropy parameter, with the distribution skewed towards the horizontal. Quantitative values are obtained by fitting Eq. (2) to radially integrated transition intensities, as a function of angle. Applying this to the inverse Abel transformed velocity-mapped image of Fig. 3 gives anisotropy parameters $\beta(X^1\Sigma_g^+) = +1.75(5)$ and $\beta(a^3\Pi_u) = -0.35(1)$.

The measured anisotropy parameters are directly related to the interference of detachment partial waves by the well-known Cooper–Zare anisotropy formula, applicable for a central potential,

$$\beta_\ell = \frac{\ell(\ell-1)\chi_{\ell,\ell-1}^2 + (\ell+1)(\ell+2)\chi_{\ell,\ell+1}^2 - 6\ell(\ell+1)\chi_{\ell,\ell+1}\chi_{\ell,\ell-1}\cos(\delta_{\ell+1}-\delta_{\ell-1})}{(2\ell+1)[\ell\chi_{\ell,\ell-1}^2 + (\ell+1)\chi_{\ell,\ell+1}^2]}, \qquad (3)$$

where $\chi_{\ell,\ell\pm1}$ is the radial transition dipole matrix element for the $\ell \pm 1$ partial wave emitted from the initial atomic orbital $\ell$ and $\delta_{\ell\pm1}$ are the associated phase shifts[36]. Qualitatively, Eq. (3) describes the variation of the anisotropy of an electron distribution, with kinetic energy $\epsilon$, for detachment from an orbital with angular momentum $\ell$. Therefore, a measurement of the anisotropy of $C_2^-$ detachment is sensitive to the character of the relevant anion molecular orbitals.

For detachment from an $s$-orbital, $\ell = 0$, Eq. (3) becomes

$$\beta_s = \frac{2\chi_{0,1}^2}{\chi_{0,1}^2} = 2. \qquad (4)$$

A strong positive anisotropy close to $+2$, such as $\beta(X^1\Sigma_g^+) = +1.75(5)$, is indicative of a dominant $s$ orbital character. Conversely, for a $p$ orbital with $\ell = 1$, Eq. (3) becomes

$$\beta_p = \frac{2A^2\epsilon^2 - 4A\epsilon\cos(\delta_1 - \delta_0)}{1 + 2A^2\epsilon^2} \qquad (5)$$

where

$$A\epsilon = \frac{\chi_{1,2}}{\chi_{1,0}} \qquad (6)$$

and where the Hanstorp coefficient $A$ is used to represent the ratio of radial matrix elements[37]. Equation (6) shows that as $\epsilon \to 0$, $\beta \to 0$. A negative $\beta$ parameter close to threshold, such as $\beta(a^3\Pi_u) = -0.35(1)$, is typical of detachment from a $p$-like orbital, as seen in similar diatomic molecules BN$^{-}$[38] and NO$^{-}$[27,39] where an electron is detached from a $\pi$-orbital.

Assuming that the relevant molecular orbitals in $C_2^-$ may be accurately described by a mixture of $s$ and $p$ character, photodetachment will be governed by the modified Cooper–Zare equation[40],

$$\beta_{sp} = \frac{2Z\epsilon + 2A\epsilon^2 - 4\epsilon\cos(\delta_2 - \delta_0)}{1/A + 2A\epsilon^2 + Z\epsilon}, \qquad (7)$$

where $Z = (1-f)B_d/(fA_d)$, for an orbital $\sqrt{1-f}|s\rangle + \sqrt{f}|p\rangle$. Parameters $A_d$ and $B_d$ from Eq. (7) represent the scaling of the different radial dipole integrals, with the ratio $B_d/A_d = 8/3$ for $2s/2p$ mixing, whereas $\cos(\delta_2 - \delta_0)$, the phase shift between the outgoing waves, is $\approx 1$[40]. This leaves the parameter $f$, associated with the fractional percentage of $s$ and $p$ character ($f = 0$ for pure $s$, $f = 1$ for pure $p$), and the Hanstorp coefficient $A$ in Eq. (7), as the only fitting parameters.

The experimental anisotropy parameters for $C_2^-$ photodetachment at 355–295 nm (to both the ground and first-excited neutral states) are plotted in Fig. 6, together with the values of Bragg et al.[26] measured at 264 nm. By fitting the anisotropies for each wavelength

to the $sp$ character model Eq. (7), values for the Hanstorp coefficients, $A$, and fractional character percentage of the detachment orbital, $f$, may be determined. This fitting process gives values of $A = 1.88(8)$ eV$^{-1}$ and $f = 0.081(9)$ for $C_2 X^1\Sigma_g^+ \leftarrow C_2^- X^2\Sigma_g^+$ detachment and values of $A = 0.86(6)$ eV$^{-1}$ and $f = 0.87(1)$ for the excited $a^3\Pi_u \leftarrow X^2\Sigma_g^+$ transition. This corresponds to a ground-state detachment orbital with $\sim 92\%$ $s$ character, whereas the excited state corresponds to detachment from an orbital with $\sim 87\%$ $p$ character.

Modified Cooper–Zare curves (Eq. (7)) are shown in Fig. 6 for different values of $sp$ orbital character percentages, with $f = 0.081$, $0.15$, $0.8$, and $0.87$. This graph highlights a general rule of $sp$ mixing, with a higher percentage of $s$ character associated with a $\beta$ close to $+2$, whereas orbitals with more $p$ character produce $\beta$ parameters close to $-1$. From comparison to the experimental data points, it can be seen that $f(X^1\Sigma_g^+) < 0.15$ and $f(a^3\Pi_u) > 0.8$, giving lower limits on the amount of $s/p$ character of each detachment orbital.

Another useful comparison is to construct Dyson orbitals $\phi^d$, which are defined as the overlap between the initial $\phi_i^{(n)}$ and final $\phi_f^{(n-1)}$ states of the molecule,

$$\phi^d = \sqrt{N} \int \phi_i^{(n)}(1, \dots , n)\phi_f^{(n-1)}(2, \dots , n)dn, \qquad (8)$$

so that the photoelectron dipole moment $D_k$ may be written in terms of the Dyson orbital $\phi^d$ and the outgoing electron plane wave $\psi_k$[41],

$$D_k \propto \langle \phi^d | e \cdot \mathbf{r} | \psi_k \rangle. \qquad (9)$$

As Dyson orbitals may be constructed using quantum chemistry software, this will provide a direct link between the ab-initio theory and the experimental results. Dyson orbitals corresponding to $C_2^-$ photodetachment to both the ground and first excited state were calculated using Q-Chem software[30] at the CCSD(T) level of theory with a 6-311 + G(4+)(3df) basis set. Anisotropy parameters were then calculated over a range of electron kinetic energies using ezDyson software[42]. As anisotropy calculations are sensitive to the asymptotic tails of the Dyson orbitals, it has been shown in previous studies that the contribution from the diffuse Gaussian orbitals may need to be adjusted to get better agreement between experiment and theory[43]. In this case it was found that increasing the contribution from the diffuse orbitals by a factor of two was required. The resulting anisotropy curves for detachment from the $3\sigma_g$ and $1\pi_u$ orbitals are also shown in Fig. 6.

Each of the bonding schemes, double, triple and quadruple, may be compared with the above limits to deduce which one best aligns with the orbital information gained from the present experimental measurements. The double-bonded structure suggested by the MO diagram in Fig. 1b predicts that the $C_2 X^1\Sigma_g^+ + e^- \leftarrow C_2^- X^2\Sigma_g^+ + h\nu$ transition involves detaching an electron out of the $3\sigma_g$ bonding orbital and the transition to form the lowest triplet state ($a^3\Pi_u$) would involve detachment from the $1\pi_u$ bonding orbital. The $3\sigma_g$ orbital possesses predominant $s$-character and the $1\pi_u$ orbital possesses significant $p$-character (Fig. 2), as confirmed by the Dyson orbital calculations. Therefore, this orbital scheme is consistent with the angular distributions measured in Fig. 3, satisfying the orbital character requirements of at least 85% $s$ and 80% $p$ character, for the singlet and triplet transitions, respectively.

The triple-bonded structure, predicted by HO theory (Fig. 1a), suggests that $C_2$ possesses $\sigma$ bond, $2\pi$ bonds, and singly occupied $sp^1$ orbitals on each of the carbons. However, as the $sp^1$ hybrid orbitals will have close to 50% $s$ and 50% $p$ character, they are not

suitable for either the singlet or triplet detachment. Multi-configurational CASSCF calculations have also been used to predict a $C_2$ bond order of 3, with a calculated orbital configuration of $KK(2\sigma_g)^2(1\pi_u)^2(1\pi_u)^2(2\sigma_u^*)^{1.6}(3\sigma_g)^{0.4}$ (Fig. 2). Detachment to the $X^1\Sigma_g^+$ ground state would most likely occur out of what is primarily a $3\sigma_g$ orbital. However, as this orbital is mixed with both the $2\sigma_u^*$ and higher lying $1\pi_g^*$ orbitals, achieving an $s$ character purity of over 85% would seem unlikely. Furthermore, the triplet transition may involve detachment from what is primarily a $2\sigma_u^*$ orbital, which, especially when orbital mixing with $3\sigma_g$ is accounted for, is unlikely to satisfy the 80% $p$ character requirement.

Likewise, the suggestion of a quadruple bond faces a similar problem. In this scenario, it is suggested there are one $\sigma$ and two $\pi$ bonds, as before, as well as a weak bond-like interaction between the singly occupied $sp^1$ hybrid orbitals on the carbons. However, even if the highest occupied molecular orbitals/lowest unoccupied molecular orbitals are not pure $50:50 sp^1$ orbitals, it is very unlikely accelerated that they would represent over 85% $s$ purity, as is required for the $C_2(X^1\Sigma_g^+) \leftarrow C_2^-(X^2\Sigma_g^+)$ transition.

It is important to note that, as an intrinsically multi-reference system, it is not possible to exclusively assign the structure of $C_2$ to any one bonding configuration. However, the anisotropy results presented here strongly suggest that the dominant contribution arises from a double bond-like configuration, with the unusual case of having $2\pi$ bonds without an accompanying $\sigma$ bond.

Ab-initio calculations give a complete multi-configuration picture, with the possibility of many different configurations contributing to the overall state. In this work the experimental photoelectron angular distribution decomposes the detachment orbital as a linear combination of atomic-like orbitals ($s$, $p$, $d$). The measurements indicate almost pure atomic character, which supports a molecular orbital configuration for the $C_2^-(X^2\Sigma_g^+)$ anion of $KK(2\sigma_g)^2(2\sigma_u^*)^2(1\pi_u)^4(3\sigma_g)^1$. The anisotropy parameters reported here are consistent with a double-bond configuration for the neutral $C_2$ molecule, with $2\pi$ bonds and no accompanying $\sigma$ bond. This suggests that although multiple configurations may contribute to the overall state, the $2\pi$ bond configuration is likely the dominant contributor to the overall bonding structure. This would also appear to be in agreement with the other experimental observations that have been made, namely the measured C–C bond length of 1.243 Å[19], which lies between the usual lengths for double and triple bonds. The fact that this bond differs from a standard C=C double bond, may be due to the unique nature of having two $\pi$ bonds, as opposed to the usual $\sigma + \pi$ bond configuration. This work suggests that triple and quadruple bond configurations, based on the hybridisation of $sp$ orbitals, will only have a small influence on the overall bonding nature, as opposed to what some calculations and theoretical approaches have suggested recently.

## Methods

**High-resolution photoelectron imaging spectrometer.** A velocity-map imaging lens, a modified version of the original concept of Eppink and Parker[44], is incorporated co-axially into a fast-beam spectrometer. Details of the apparatus are given in refs. [39,45]. Dicarbon anions $C_2^-$ are produced by passing pure ethylene gas ($C_2H_4$) through a pulsed-valve nozzle at a stagnation pressure of $\sim 2$ atm, with supersonic expansion through a pulsed high-voltage discharge. Negative ions are extracted, accelerated to 500 eV and focused into an ion gating, bunching and potential re-referencing unit[46]. Anion mass separate travelling over a 2 m time-of-flight tube, with each mass packet bunching to an $\sim 2$ mm$^3$ volume at the interaction region, where an electrostatic gate selects the mass packet of interest. This ion packet is crossed with a detachment laser beam, generated from a Continuum Powerlite 9010 Nd:YAG laser, operated at its third harmonic, 355 nm. The laser produces between 20 and 50 mJ per pulse at 10 Hz, measured near the interaction

region. A Continuum Sunlite EX OPO combined with an FX-Doubler was used to produce wavelengths at 295, 300, 305, 315 and 325 nm, with laser powers of 2–10 mJ. The precise wavelength of the laser light is measured using a wavemeter (HighFinesse WS7 UV), giving calibrated wavelengths of 289.612(1), 299.593(1), 304.596(1), 314.581(1), 324.548(1) and 354.8071(9) nm.

Photodetached electrons are velocity mapped to a 75 mm diameter MCP/phosphor screen detector (Burle), which is imaged by a 2048 × 2048 pixel monochrome CCD camera (PCO 2000). Each camera frame is transferred to a computer at a 10 Hz repetition rate and is processed in real time to identify individual electron events, centroiding position to a sub-pixel accuracy. The electron positions are written to a data file for subsequent analysis.

**Image analysis**. An image of the velocity-mapped photodetached electrons at the detector is then obtained through binning of the centroided electron-event positions into a rectangular pixel-grid image, of arbitrary pixel number, based on signal-to-noise ratio. The velocity-map image is centred and circularised by an angular-dependent radial scaling determined by comparing adjacent radial slice intensity profiles[47]. An inverse Abel transformation of the VMI, based on the algorithm of Hansen and Law[48,49], returns a slice image of the 3D electron source distribution.

Absolute energy calibration of the photoelectron spectra is achieved using published measurements of species, including $O^{-}$[45] and $O_2^{-}$[50], which have been studied under similar conditions as used for the $C_2^{-}$ measurements.

## Data availability

All data supporting the findings of this study are available within the Article and from the corresponding author upon reasonable request. Experimental data are available at https://physics.anu.edu.au/research/portal/vmi.

## Code availability

All of the computer code supporting the findings of this study are available from the corresponding author upon reasonable request. The code used in the velocity-map image analysis is available at https://github.com/PyAbel/PyAbel.

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

## Acknowledgements

This research was supported by the Australia Research Council Discovery Project Grant DP160102585.

## Author contributions

B.A.L. performed the experiments and ab-initio calculations. B.A.L. and S.T.G. were involved in the data analysis, the experiment design, and wrote the computer codes used to examine the experimental data. B.A.L., S.T.G., B.R.L. and R.W.F. interpreted the results in relation to bonding in $C_2$. B.A.L. wrote the manuscript, with input, comments and discussion from S.T.G., B.R.L. and R.W.F.

## Competing interests

The authors declare no competing interests.
