## [Peer Review File · Nature Communications]

REVIEWERS' COMMENTS:

Reviewer #2 (Remarks to the Author):

I would like to commend the authors for taking on board the suggestion to acquire the additional data. I appreciate it is never something one wants to hear, but I do feel that the case is now much stronger based on the experimental PADs (i.e. it now has an energy dependence). I think this paper presents an interesting and important contribution to the discussion about the C2 bond and offers concrete experimental insight into its bonding.

There is one v. minor point. On page 5 col 2, the authors explain how they get quantitative anisotropy parameters and they say they fit the radially-integrated signal as a function of angle from Fig. 3. They should be a little more explicit and say that they use the central slice of the Abel inverted image (which I am assuming they did because otherwise, they'd get a different answer!).

Below is our response to the comments made by Reviewer #2:

Reviewer #2 -- NCOMMS-19-28690-T/Laws

I would like to commend the authors for taking on board the suggestion to acquire the additional data. I appreciate it is never something one wants to hear, but I do feel that the case is now much stronger based on the experimental PADs (i.e. it now has an energy dependence). I think this paper presents an interesting and important contribution to the discussion about the C2 bond and offers concrete experimental insight into its bonding.

There is one v. minor point. On page 5 col 2, the authors explain how they get quantitative anisotropy parameters and they say they fit the radially-integrated signal as a function of angle from Fig. 3. They should be a little more explicit and say that they use the central slice of the Abel inverted image (which I am assuming they did because otherwise, they'd get a different answer!).

This sentence has been fixed to clarify that the anisotropy parameters are calculated from the Abel inverted image, not the raw data. It now reads "Applying this to the inverse Abel transformed velocity-mapped image of Fig. 3 gives anisotropy parameters $\beta(X^1\Sigma_g^+)= +1.75(5)$, and $\beta(a^3\Pi_u)= -0.35(1)$."